# Dissipative Particle Dynamics Simulation for the Self-Assembly of Symmetric Pentablock Terpolymers Melts under 1D Confinements

**DOI:** 10.3390/polym15193982

**Published:** 2023-10-03

**Authors:** Yingying Guo, Linqing Bai

**Affiliations:** 1School of Science, Qingdao University of Technology, Qingdao 266525, China; 2School of Mechanical and Automotive Engineering, Qingdao University of Technology, Qingdao 266525, China; skdbailinqing@163.com

**Keywords:** symmetric pentablock terpolymres, phase behavior, DPD simulation

## Abstract

The phase behavior of CBABC pentablock terpolymers confined in thin films is investigated using the Dissipative Particle Dynamic method. Phase diagrams are constructed and used to reveal how chain length (i-block length), block composition and wall selectivity influence the self-assembly structures. Under neutral walls, four categories of morphologies, i.e., perpendicular lamellae, core–shell types of microstructures, complex networks, and half-domain morphologies, are identified with the change in i-block length. Ordered structures are more common at weak polymer–polymer interaction strengths. For polymers of a consistent chain length, when one of the three components has a relatively smaller length, the morphologies transition is sensitive to block composition. With selective walls, parallel lamellae structures are prevalent. Wall selectivity also impacts chain conformations. While a large portion of chains form loop conformations under A-selective walls, more chains adopt bridge conformation when the wall prefers C-blocks. These findings offer insights for designing nanopatterns using symmetric pentablock terpolymers.

## 1. Introduction

Block copolymers (BCPs) are polymers comprising of spatial arrangements of different types of blocks. One of the reasons that these polymers have attracted significant scientific attention is their self-assembly ability. Due to the thermodynamic incompatibility among blocks, BCPs melts could phase separately into abundant, well-defined periodic nanostructures with critical dimensions between 5 and 100 nm [1] and with specific orientation. In addition, the self-assembly ability of polymers helps to achieve structures with improved mechanical-elastic properties [2].

The self-assembly of BCPs melts in thin films has been regarded as one of the most popular bottom-up techniques for manufacturing nanodevices such as sensors, solar cells, ultrafiltration membranes, mask in nanolithography, and photonic and electronics nanodevices [3,4,5,6,7,8,9,10] due to the low cost, simple processing, high efficiency, and excellent scalability in obtaining nanopatterns [11,12]. For the BCPs melts film, compared with bulk, introducing the confinement brings extra parameters for controlling the phase behaviors of polymers. Those include the commensurability between the film thickness and characteristic length of BCP nanostructures, as well as the interactions between the polymer and the substrate or the upper bounding surface [13]. Thus, the phase behavior of BCPs melts in film differs from that in bulk. The BCPs melts self-assembly in thin films has been extensively studied experimentally [14,15,16,17,18,19,20] and theoretically [21,22,23,24,25,26,27] to understand the different phase behaviors from those in bulk. Aviv et al. [16] studied the self-assembly of bottlebrush block copolymers melts on different types of substrates with both experiment and computational simulation. A nonlamellar morphology was first demonstrated for a symmetric bottlebrush block copolymer. It suggested that a deposited film initially responding to the substrate selectivity may give rise to unexpected transient morphologies under self-assembly. In simulation, a common and easy way to model the thin films is by placing the polymer under the confinement of two impenetrable surfaces. Li et al. [25] constructed a phase diagram of diblock copolymers melts confined between two flat surfaces with two identical preferential surfaces using SCFT. Compared with the bulk phase diagram, the microphases were enriched significantly. Around 20 morphologies were observed with respect to the volume fraction and the film thickness. Four categories of ordered phases, sphere, cylinder, perforated lamellae, and lamellae, were classified. Jiang et al. [26] studied the phase behavior of ABC triblock terpolymers thin films directed by polymer brushes with SCFT. By varying the block composition, ordered complex morphologies such as parallel lamellar phase with hexagonally packed pores at surfaces, perpendicular lamellar phase with cylinders at the interface, and perpendicular hexagonally packed cylinders phase with rings at the interface were observed with the fixed film thickness and the brush density. Lamellar phases or cylindrical phases with desired directions could also be obtained. Liu et al. [27] examined the phase behavior of ABC star terpolymers confined between two identical parallel surfaces using a simulated annealing method. The results showed that the orientation of a confined phase depends on the “effective surface preference”, which is a combined effect of the interfacial interaction strength ratio, the surface preference, and the entropic preference.

Given that polymer properties are closely linked to the molecular structure and monomer distribution along the constituent chains, it seems that a careful consideration of the length of each block, the number and order of blocks, and the inclusion of monomers with specific functional groups provides endless opportunities for fine-tuning the properties of the self-assembled nanostructures [28]. Among various types of BCPs, multiblock copolymers are envisioned as promising materials with enhanced properties and functionality compared with their diblock/triblock counterparts [29]. The symmetric ABCBA linear pentablock terpolymer with a block number of 5 has been regarded as a representative model and starting point for investigating the self-assembly of multiblock copolymers [30]. The phase behavior of ABCBA linear pentablock terpolymers in bulk and in solution has been observed extensively both in experiment [31,32,33,34,35,36,37] and in computational simulations [38,39,40,41]. For bulk states, diverse continuous network structures were widely obtained. Bates et al. [34] further concluded that the O70 microphase structure self-assembled from non-frustrated poly(ethylene oxide)-b-polystyrene-b-polyisoprene-b-polystyrene-b-poly(ethyleneoxide) OSISO pentablock terpolymers had better tensile properties compared with that of OSI triblock terpolymers. This is ascribing to the intrinsic topological structures of the ABCBA, which make it possible for the polymer chain to have loop conformation in addition to a linear arrangement. However, to the best of our knowledge, the phase behavior of symmetric pentablock terpolymers under confinement has rarely been reported.

In the present work, the phase behavior of symmetric CBABC pentablock terpolymers melts has been explored with the DPD simulation method. We specifically examined the influence of surface selectivity on polymer self-assembly. For the nonselective wall, we studied the effects of both chain length and block composition. Phase diagrams were constructed to systematically discuss the self-assembly behavior of the linear pentablock terpolymers by considering the variation of i-block length and composition ($f_{A}$, $f_{B}$, and $f_{C}$). For the selective wall, phase diagrams were built to map out the relationship between i-block length and the surface preference (toward the A- or to the C-blocks) under a fixed interfacial interaction strength. Our findings provide a comprehensive understanding of phase behavior exhibited by symmetric pentablock terpolymers melts when confined within thin films.

## 2. Materials and Methods

The self-assembly behavior of symmetric CBABC pentablock terpolymers melts under two parallel walls was monitored using the dissipative particle dynamics (DPD) method. DPD is a particle-based mesoscopic simulation technique first introduced by Hoogerbrugge and Koelman [42]. In the simulations, polymer chains are coarse-grained into chains composed of DPD beads, all of which are of equal size.

In DPD, the motion of each bead is governed by Newton’ s second law of motion. The total force acting on each bead i is ${\vec{f}}_{i}$ and consists of four components. Pairwise components, i.e., the conservative force ${\vec{F}}_{ij}^{C}$, the dissipative force ${\vec{F}}_{ij}^{D}$, and the random force ${\vec{F}}_{ij}^{R}$, become effective when the distance between two beads i and j is within the cut-off radius $r_{c}$. The conservative force ${\vec{F}}_{ij}^{C}$ is a soft-repulsive force and is given by
(1)$${\vec{F}}_{ij}^{C}=\left\{\begin{array}{c} a_{ij}\left(1-r_{ij}/r_{c}\right){\hat{r}}_{ij}\left(r_{ij}<r_{c}\right)\\ 0\;\left(r_{ij}\geq r_{c}\right) \end{array}\right.$$
where ${\vec{r}}_{ij}={\vec{r}}_{i}-{\vec{r}}_{j}$, $r_{ij}=\left|{\vec{r}}_{ij}\right|$, ${\hat{r}}_{ij}={\vec{r}}_{ij}/\left|{\vec{r}}_{ij}\right|$. $a_{ij}$ is the maximum repulsion between beads i and j, and $r_{c}$ is the cut-off radius with value 1.0. A spring force ${\vec{F}}_{ij}^{S}$ is introduced between beads connected by covalent bonds to simulate polymer chains. It follows a simple harmonic potential with a spring constant *k* = 8.0.

The model systems we are examining are composed of CBABC pentablock terpolymers, which consist of A-, B-, and C- beads and are confined by walls (W). Figure 1 provides a schematic representation of the molecular structure of the linear pentablock terpolymers. The term “symmetric” means that the sequence of the blocks is mirrored about the center block. In the figure, the A-, B-, and C- blocks are color-coded as yellow, green, and red, respectively. We utilized two identical, rigid walls (made of W-beads) at the top and bottom z boundaries. Meanwhile, periodic boundary conditions were applied along the x and y directions. The walls were built with a face-centered cubic structure with the (100) plane facing the melt, and have a lattice spacing of 0.855$r_{c}$.

The simulations begin with random distributed pentablock terpolymers in a box of size $ar_{c}\times ar_{c}\times hr_{c}$, where a depends on the length of the chain. The polymer chain length is characterized as the total number of beads N. For N≤30, box size a=60. For N=36 and 42, the size a is set to 80 and 84, respectively. h is the film thickness between two rigid walls. The value is fixed at $6.4r_{c}$, which corresponds to an ultrathin film compared with x and y (a≥60) dimensions. Given the huge phase space of the symmetric CBABC pentablock terpolymers system under confinement, we have made some simplifications. All interaction parameters among polymer beads were set to be equal, i.e., $a_{AB}=a_{BC}=a_{AC}$. For a neutral wall, the interaction strengths between all blocks and the wall were fixed at 120 (i.e., all blocks are wall-repelling), while for a selective wall the interaction strength between the selective block and the wall was reduced to 25, while the interactions involving the other two blocks remained at 120 (i.e., the selective block has wall affinity while other blocks are wall-repelling). To study the impact of chain length on the morphologies, we have kept the length of two blocks constant while adjusting the length of the remaining block. As the chain length is varied based on the individual length of the three components, we will use i-block length in subsequent discussions. All computations were equilibrated for a minimum of 500,000 timesteps, with the timestep being Δt=0.04.

## 3. Results and Discussion

The selectivity of a confining surface plays an important role in determining the self-assembly morphologies of polymers. In this paper, we will explore the phase behaviors of the pentablock terpolymer melts under two distinct scenarios: (1) the wall exhibiting no preference towards any of the blocks (neutral wall), and (2) the wall demonstrating a preference, specifically for A- or C-blocks (selective wall). For scenario 1, our discussion will revolve around the influence of i-block length and block compostion on the self-assembly morphologies. For scenario 2, our emphasis will be on understanding the impact of chain length when the walls are selectively inclined towards A- or C-blocks.

### 3.1. Neutral Wall

#### 3.1.1. Influence of the i
-Block Length

In this section, we modify the length of polymer chains by adjusting the length of one block while maintaining the lengths of the other two blocks at a consistent size of 6. The lengths of the variable blocks are set at 2, 6, 12, 18, 24, and 30 (corresponding to composition of 0.14, 0.34, 0.50, 0.60, 0.68, 0.72, respectively). This results in overall chain lengths of 14, 18, 24, 30, 36, and 42. The interaction parameters between all blocks and the wall are set at 120, indicative of a strongly repulsive wall. Appendix A presents the phase triangle at different polymer–polymer segregation strengths.

##### Phase Behavior

We firstly examined the phase transition with $f_{C}$ along the situation of $f_{A}=f_{B}$. Detailed morphologies at different polymer–polymer interaction energies are depicted in Figure 2. At $f_{C}=0.14,$ a disordered phase was observed at a weaker polymer–polymer segregation strength ($a_{AB}=a_{BC}=a_{AC}=40$). As the block immiscibility increased ($a_{AB}=a_{BC}=a_{AC}=80$), the phase separation intensified, leading to the formation of a parallel lamellae structure (LAM3_⫽_). In this structure, all layers were oriented parallel to the confining wall. The LAM3_⫽_ structure comprised a central layer of aggregated A-blocks, which was sandwiched between two layers formed by B-blocks, while the minority C-blocks were situated at the interface between the polymer and the wall, as shown in Appendix A. As $f_{C}$ rose to 0.34, with the system corresponding to an equal composition of all blocks, the morphology transitioned to perpendicular lamellae at $a_{AB}=a_{BC}=a_{AC}=40$ and to fingerprint lamellae at $a_{AB}=a_{BC}=a_{AC}=80$. Unlike LAM3_⫽_, the lamellae were then oriented perpendicular to the confined wall. For $f_{C}\geq 0.5$, where C-blocks constituted the majority of the chain, molecules assembled into a series of core–shell structures embedded in a C-matrix. In these structures, B-blocks shaped the shell while A-blocks formed the core. Such core–shell structures have also been observed in symmetric pentablock terpolymers in bulk when the end-block composition was high [40]. At a weaker polymer–polymer interaction strength, the core–shell structure underwent a transition sequence: core–shell cylindrical structure (CSC), → core–shell irregular structure (CSI), → core–shell spherical phase (CSS). At stronger interaction strength, the sequence was core–shell cylinder (CSC) → core–shell bead-string (CSBS) → core–shell spherical and short cylindrical phase (CSSC). C-rich perforated lamellae could be observed when $f_{C}>0.6$, irrespective of the polymer–polymer interaction strength. In summary, within the parameter space we have explored, the morphologies could be broadly categorized into three primary classes in terms of the phase behavior of the C-component: (1) lamellae, (2) cylinders, and (3) perforated lamellae. Each class contains several kinds of related structures.

Next, we moved to the system where $f_{B}=f_{C}$. At an interaction strength of $a_{ij}=40$ (i,j=A, B, C), a sequence of transition (Appendix A) was observed, moving from a disordered phase (D) to perpendicular lamellae (LAM3_⊥_), then to the core–shell cylinder phase (CSC), and finally to a core–shell irregular structure (CSI) as $f_{A}$ varied. The phase diagram, shown in Appendix A, does not exhibit A–C reflection symmetry. The different points are located at ($f_{A},\;f_{B},\;f_{C})\in$ {(0.68, 0.16, 0.16), (0.72, 0.14, 0.14)}, where C-cores show an irregular structure with several protrusions (Appendix A). This irregularity results from the decreased interaction energies among the blocks, causing the building blocks to preferentially align with each other and allowing the chains to stretch freely. Since the C-block was located at both free ends of the chain, the cores formed by C-blocks exhibited more protrusions compared to those formed by A-blocks. When the polymer–polymer interaction strength was increased to 80, the distinct point for A–C reflection symmetry was found at a smaller $f_{A}$ ($f_{A}=0.14$). This asymmetry could be ascribed to the specific chain architecture, i.e., the A-block is connected to two B-blocks and lacks free ends, which is different from C-blocks. This results in different contributions of C-block and A-block during micro-phase separation when $f_{A}$ and $f_{C}$ are small [40]. Despite these subtle structural differences, the morphologies could also be divided into three primary classes based on the phase behavior of component A: (1) lamellae, (2) cylinders, and (3) perforated lamellae. These classes align with those identified for $f_{C}$.

Figure 3 presents the morphology transition with $f_{B}$ when $f_{A}=f_{C}$. At $f_{B}=0.14$ and $a_{AB}=a_{BC}=a_{AC}=40$, a disordered phase was obtained. As $f_{B}$ increased to 0.33 and 0.5, a perpendicular lamellae structure (LAM3_⊥_) appeared. Continuing to elevate $f_{B}$ from 0.5 resulted in the majority B-blocks forming a dense matrix layer. Within this matrix, the minority A and C blocks manifested as small, irregular segments (Figure 3a). When the segregation strength intensified to $a_{AB}=a_{BC}=a_{AC}=80$, the molecules tended to pack into ordered A-, B-, and C-rich microdomains (Figure 3b). Fingerprint lamellae structures were obtained at $f_{B}\leq 0.5$. For $0.5<f_{B}\leq 0.68$, a double gyriod (DG) phase was observed. The double-gyroid (DG) structure (Appendix A) is characterized by two independent, opposite-handed, interpenetrating networks formed by A- and C-blocks. Diamond, hexagonal packed cylinders, and spherical phase, typically observed in the bulk [38], have disappeared. This absence might be attributed to the higher degree of packing frustration of those morphologies compared with the gyroid phase [38]. When $f_{B}$ increased to 0.72, the gyroid structure was broken, resulting in a network structure with interpenetrate A- and C-domains.

Based on the aforementioned information, our phase diagram with respect to i-block length suggests that the morphology transitions associated with A-blocks and C-blocks are notably similar. Regardless of the block–block interaction strength, the morphologies could be categorized into lamellae, cylinders, and i-rich perforated lamellae, in terms of the phase behavior of the i-component (i=A, C). The morphology transition with the B-block length is influenced by the interaction between blocks, especially when the fraction of the B-block is high. At high $f_{B}$ and strong block–block interaction strengths, a double gyroid phase can be observed. Additionally, the lamellae structures with parallel A/B/C rich domains are more easily obtained at low block–block interaction strengths and equal A/B/C composition.

##### Characterization of C-Rich Perforated Lamellae

The perforated lamellae phase of block copolymers is an interesting structure in which the minority components impart three-dimensional continuity to the majority components [43]. Such a structure holds great potential for developing novel materials with unique porous nanostructures, i.e., next-generation ultrafiltration (UF) membranes featuring intelligent nanochannels [44]. The i-rich perforated lamellae appear at high $f_{A}$ or $f_{C}$. Given the analogous morphology transition with $f_{A}$ and $f_{C}$, this section will primarily focus on C-rich perforated lamellae.

To compared the structural differences of the C-rich perforated lamellae obtained from different segregation strengths, we calculated the size of the pores and the radial distribution function at grid point ($f_{A},\;f_{B},\;f_{C})\in$ {(0.16, 0.16, 0.68), (0.14, 0.14, 0.72)} (Figure 4). The size of the matrix pores was evaluated from A-cores. The size of A-cores is defined by their average radii $r_{a}=\sqrt{\frac{1}{3}(I_{1}+I_{2}+I_{3})}$, where $I_{1}$, $I_{2}$ and $I_{3}$ are the eigen values of their moment of inertia. Figure 4a,b show that the number of pores reduces with $a_{ij}$ (i,j=A, B, C). For grid point ($f_{A},\;f_{B},\;f_{C})\in$ (0.16, 0.16, 0.68), the mean value of the pore size was about 1.86 with a standard deviation of 0.06 at a weak interaction strength, while for a strong interaction strength the mean value of the pore size was about 2.45 with a standard deviation of 0.26. As $f_{C}$ increased to 0.72, the mean value of the pore size was about 2.10 with a standard deviation of 0.12 at a weak interaction strength, while about 2.36 with a standard deviation of 0.22 at a strong interaction strength (Figure 4b). This analysis reveals that pores formed under a weak interaction energy exhibit a more uniformly distribution. Figure 4c,d present the radial distribution function, g(r), between components C and A under different polymer–polymer immiscibility and C-block length. For all the cases, the g(r) exhibited several pronounced peaks, indicative of the alternating presence of the A block domains. The number of peaks decreased with the C-block length at a weak interaction energy, suggesting a decline in ordering. Conversely, peak numbers rose with C-block length at a stronger interaction strength. That implies that the order decreased with C-block length at weaker polymer–polymer interaction energies but enhanced at stronger interaction energies.

#### 3.1.2. Influence of the Block Composition

In this section, we explore the effect of composition on the morphology transition under confinement. The interaction parameters among blocks were set to $a_{AB}=a_{BC}=a_{AC}=80$. Within the intermediate region of the phase triangle (Figure 5), i.e., ($f_{A},\;f_{B},\;f_{C})\in${(0.34, 0.33, 0.33), (0.44, 0.22, 0.34), (0.22, 0.44, 0.34), (0.34, 0.22, 0.44), (0.22, 0.34, 0.44), (0.34, 0.44, 0.22), (0.44, 0.34, 0.22)}, where the compositions of the three components are comparable, molecules aggregated into lamellae structures. The phase behaviors in these areas of the phase diagram still resembled the phase sequence observed in bulk [40].

Our attention then shifted to morphology transitions when one of the three compositions was relatively small. The morphology transition with $f_{B}$, given $f_{C}=0.11$, is presented in Figure 6a. For $f_{B}\leq 0.22$, half-domain morphologies were obtained. In these structures, the majority A-blocks formed a single layer parallel to the wall, while the minority B- and C- blocks formed nanodomains that decorated the two sides of the layer. These nanodomains manifested as irregular patterns at $f_{B}=0.11$ (Figure 7a) and as half core–shell cylinders (Figure 7b) at $f_{B}=0.22$. Such half-domain morphologies have also been reported for diblock copolymers under ultra-thin confinement [25]. For ${0.22<f}_{B}\leq 0.56$, a series of C-core/B-shell structures, including core–shell network and core–shell cylinder structures, were observed. A single gyroid structure formed by A-blocks appeared at $f_{B}=0.67.$ At $f_{B}=0.78$, B-blocks coalesced to a dense layer, while A-blocks formed short clusters within the layer.

For the situation with low A-composition ($f_{A}=0.11$), the phase transition sequence with $f_{B}$ was as follows: double layer half core–shell cylinder (HCSC) → core–shell spherical/cylinder structure (CSSC) → core–shell thread structure (CST) → core–shell cylinder (CSC) → double gyroid (DG) → dense layer (DL). For the CST structure observed at $f_{B}=0.33$, the A and B blocks formed B-shell/A-core strings threading through the C-matrix (Figure 7c). We further examined the morphology transition at $f_{B}=0.11$ (Appendix A). At a medium $f_{A}$ ($f_{C}$), fingerprint lamellae structures were investigated. Morphologies transitioned to double layer half-domain structures at higher $f_{A}$ ($f_{C}$) (=0.78). As $f_{A}$ ($f_{C}$) decreased to 0.67, core–shell structures were obtained.

The phase diagram, in relation to the i-block composition with a fixed polymer chain length, suggests that the phase behavior is almost analogous to bulk phase transition when the compositions of all blocks are comparable, resulting in lamellae structures. In the case with low $f_{i}$ (i=A,C), morphologies changed sensitively with other two blocks. A series of core–shell structures were obtained. The i-rich (i=A or C) perforated lamellae formed at points (0.56, 0.33, 0.11), (0.11, 0.33, 0.56) and (0.11, 0.22, 0.67) showed less structural order compared with those obtained at varied polymer chain lengths.

### 3.2. Selective Wall

Unlike neutral walls, selective walls have a profound impact on the morphology of self-assembled structures. Selective walls can guide the orientation of the structures. In addition, the presence of selective walls often results in the formation of specific morphologies at the wall–polymer interface. This is of great importance in applications where the morphology of the polymer assembly directly impacts its functionality. In this section, we focus on the wall with a preference for either A- or C-blocks. To achieve this selectivity, the interaction energy between the preferred block and the wall was set to 25, while the interaction energies between other blocks and wall were set at 120. This was corresponding to a scenario in which the wall strongly attracts one block and repels the other two. The interaction energies among the blocks remained at 40. We then explored the effect of i-block length, as we did in Section 3.1.1.

#### 3.2.1. Influence of A(C)-Block Length

Figure 8a illustrates the morphology transition based on the length of block C under both the A-selective and C-selective walls. The molecules self-assembled into various lamellae structures, except for $l_{C}=30$. At $l_{C}=30$, a single layer with patterns (SLP) formed by A- and B-blocks was observed under the A-selective wall. In contrast, under the C-selective wall, core–shell morphologies (CS) such as core–shell cylinder, ring, and spherical structures were observed.

For $l_{C}<30$, the density number distribution profiles, which help in understanding the lamellae structures, were plotted along the z-direction and given in Appendix A. Regardless of the C-block length, layered structures with selective blocks positioned at the polymer–wall interface were consistently observed. Under the A-selective wall, a center layer comprising mixed B- and C-blocks (ML) was observed at $l_{C}=2$ (Appendix A(a1)). As $l_{C}$ increased, B-blocks and C-blocks segregated into three distinct layers, with the C-layer sandwiched between two B-layers (LAM_⫽_) (Appendix A(a2–a5)). The changing density profiles of B-blocks with the length of the C-block are presented in Figure 8b, revealing an expanding distance between the two B-peaks with $l_{C}$. This suggests that the thickness of the center layer of the lamellae grows with the length of the C-blocks.

Figure 8d,f display the chain conformation profiles as a function of $l_{C}$. At small $l_{C}$, both the parallel and perpendicular components of $R_{EE}^{2}$ remained low. The perpendicular components of $R_{g}^{2}$ and $R_{EE}^{2}$ exhibited minimal growth with increasing $l_{C}$. The increases in $R_{g}^{2}$ and $R_{EE}^{2}$ were mainly dependent on their parallel components as $l_{C}$ grows, indicating that polymer chains primarily stretch within the xy plane. The perpendicular component of $R_{EE}^{2}$ of chains and their distributions for various $l_{C}$ are detailed in Figure 9a. A large portion of chains had an $R_{EE,⊥}^{2}$ value less than 5, suggesting that most chains have their two end C-beads positioned closely in the z direction. This can be attributed to the chain topology, where A-beads are centrally located in the chain and C-beads are at the ends. If the wall favors A-blocks and strongly repels C-blocks, the chain will fold into a loop conformation, with the middle-block in contact with the wall and the two end-blocks situated centrally in the film.

In the case of a C-selective wall, a morphology transition from parallel lamellae (LAM_⫽_) to mixed layer (ML) and then to perforated lamella (PL) was observed. The parallel lamellae structures had an A-layer sandwiched between two B-layers at small $l_{C}$. As $l_{C}$ grew, A- and B-blocks gradually mixed. The thickness of the center layer of lamellae reduced as $l_{C}$ increased from 2 to 12 (Figure 8c), which contrasted with the A-selective wall, where the thickness grew with $l_{C}$. At $l_{C}=12$, the center layer thickness reached its minimum. Further growth in the C-block length ($l_{C}$ = 18, 24) resulted in AB-mixed perforated layer structures. In these structures, portions of C-blocks formed 3D continuity that embedded to the perforations of the layer. Similar to the scenario with the A-selective wall, chains increasingly stretched in the parallel plane with $l_{C}$ (Figure 8e,g). Compared to the A-selective wall, the perpendicular component of $R_{EE}^{2}$ was notably larger under the C-selective wall. That is due to the positioning of C-blocks at the free end of the chain. If the wall favors the two end C-blocks and strongly repels A-blocks, parts of the chain will extend along z direction, with both ends in contact with the top and bottom walls, resulting in a bridge conformation. This hypothesis is supported by the probability distribution function of $R_{EE,⊥}^{2}$ (Figure 9b), where the fraction of chains (number of chains) with a mean square end-to-end distance along the z direction exceeding 25 increased under the C-selective wall (Figure 9b). We further observed that the fraction of large $R_{EE,⊥}^{2}$ decreased with increasing $l_{C}$. This trend occurred because, as $l_{C}$ grows, more C-beads per chain have the opportunity to come into contact with the wall. This leads to a reduction in the number of chains with both ends in contact with the wall.

We subsequently examined the morphology transition based on the length of the A-block under both A-selective and C-selective conditions (Appendix A). Compared to the variations observed with $l_{C}$, the phase diagram indicated a swap in morphology transition related to wall selectivity (Figure 8a). For the A-selective scenario, we identified a phase transition sequence from parallel lamellae (LAM_⫽_) to mixed layer (ML), to perforated lamellae (PL), and finally to core–shell structure (CS). The density number distribution function (Appendix A) for B-blocks indicated that the thinnest center layer was found at $l_{A}=18$, compared with $l_{C}=12$ when $l_{C}$ was varied under the C-selective wall. This can be attributed to the wall affinity for A-blocks, which makes the chain fold from the center, thereby conserving more space. The perpendicular components of $R_{g}^{2}$ and $R_{EE}^{2}$ (Appendix A) remained at a low value and almost unchanged, while their parallel components increased with $l_{A}$.

In the case of the C-selective wall, the morphology transitions observed were from the mixed layer (ML) to the parallel lamellae (LAM_⫽_), and then to a single layer with patterns (SLP). The density profile suggests that the thickness of center layer grows with $l_{A}$ (Appendix A), which is similar to Figure 8c. The perpendicular component of $R_{EE}^{2}$ (Appendix A) for small $l_{A}$ ($l_{A}=1$) displayed a significant deviation when compared to $l_{C}=1$ (Figure 8g). For $l_{A}=1$, both end-blocks had three beads each. However, for $l_{C}=1$, each end-block contained just a single bead. As a result, in the latter case, many C-beads came into contact with the wall due to wall selectivity, leading to a larger $R_{EE,⊥}^{2}$.

#### 3.2.2. Influence of B-Block Length

We further investigated the morphology transition in relation to the length of the B-block under different wall selectivity values. The morphology transition, based on the length of B-block, exhibited similar phase behaviors under both A-selective and C-selective walls (Figure 10a). In both cases, at $l_{B}=2$, the molecules self-assembled into a lamella with a center layer composed of all blocks (Appendix A(a1,b1)), where C(A) blocks predominantly aggregated within the center layer. As $l_{B}$ increased to 6, the A-, B- and C- blocks began to phase separate, forming A-rich, B-rich, and C-rich parallel layers (Appendix A(a2,b2)). For $l_{B}>6$, B-beads gradually penetrated the C(A)-rich domain, resulting in a mixed center layer of C(A)- and B-beads. Contrasting with the scenarios where $l_{A}$ and $l_{C}$ changed, the thickness of the lamellae remained almost unchanged with $l_{B}$ (Figure 10b,c). The end-to-end distance for the C-selective wall was consistently larger than that of the A-selective wall with increasing $l_{B}\;(l_{B}<30)$ (Figure 10d–f). When $l_{B}$ was further increased to 30, a distinct single layer with patterns (SLP) emerged. These patterns consisted of independent nanodomains formed by A- and C-blocks, which were distinct from the core–shell patterns observed with changing $l_{C}$ ($l_{A}$).

Overall, when considering the phase behavior influenced by chain length and wall selectivity, we note that lamellae oriented parallel to the wall could be obtained at $f_{C}$ ($f_{A}$) = 0.34~0.68 under the A(C) selective wall. This observation contrasts with the behavior of lamellae formed under nonselective walls, which orient perpendicular to the wall and typically form when the fractions of all blocks are comparable. The thickness of the center layer of the lamellae could be adjusted by the length of block C(A). For larger values of $f_{C}(f_{A}$), BA(C)-mixed perforated lamellae were obtained under the C(A) selective wall, while C(A)-rich perforated lamellae formed under the nonselective wall. We also found that chain conformations of the pentablock terpolymers under confinement could be adjusted through wall selectivity. Under the C-selective wall some chains adopted a bridge conformation, while under the A-selective wall a large population of chains adopted a loop conformation. Given that the mechanical properties of the self-assembly structure could be influenced by chain conformation [34], researchers have the option to modify chain conformation by altering wall selectivity.

In summary, considering the influence of wall properties on self-assembly morphologies, we observed five types of morphologies that were particularly relevant to the application field. We have included a table (Table 1) that outlines these structures and their formation conditions. This table could serve as a guideline for comparing different self-assembly behaviors of different type of polymers and for synthesizing corresponding nanopatterns.

## 4. Conclusions

In this study, we systematically examined the self-assembly behavior of CBABC pentablock terpolymers between two parallel walls using the DPD method. Two scenarios based on wall properties have been discussed.

For the neutral wall, phase diagrams were constructed considering chain length (i-block length, where i=A, B, C) and block composition. Perpendicular lamellae structures were observed when the fractions of A-, B-, and C-blocks were comparable. The phase diagram, in relation to A (or C)-block length, indicates that molecules self-assemble into A (or C)-rich perforated lamellae with a higher $f_{A}$ or $f_{C}$, regardless of block–block interaction strength. Notably, the i-rich (i=A, C) perforated lamellae exhibited uniformly distributed pore sizes under conditions of weak polymer–polymer interaction strength. In contrast, the phase diagram related to the B-block length showed that a double gyroid phase formed at high $f_{B}$ with a strong block–block interaction strength.

For the selective wall, the lamellae structure has been identified as the dominant phase under A (or C)-selective wall. In contrast to the perpendicular alignment of the lamellae observed under neutral walls, the lamellae under selective walls aligned parallel to the wall. The thickness of the center layer of the lamellae could be modulated by adjusting the length of the A (or C)-block. Upon further examination of the chain conformation, we found that the wall selectivity influenced chain conformation. While the loop conformation was prevalent under A-selective walls, there was a tendency for more chains to adopt the bridge conformation under C-selective walls.

In short, the orientation of the lamellae structures could be controlled by modifying wall properties. Perforated lamellae could be obtained when the fraction of either the two end blocks or the middle blocks was high. To tune the mechanical properties of the self-assembled morphologies of pentablock terpolymers, one could adjust the wall selectivity to either the two free end blocks or the center blocks. This modulation enables polymer chains to exhibit varying ratios of bridging to looping chains.

## Figures and Tables

**Figure 1 polymers-15-03982-f001:**
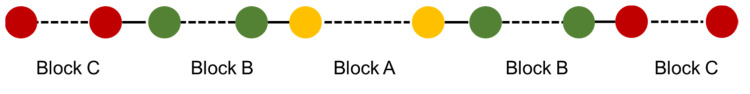
Schematic showing the structure of linear symmetric CBABC pentablock terpolymers.

**Figure 2 polymers-15-03982-f002:**
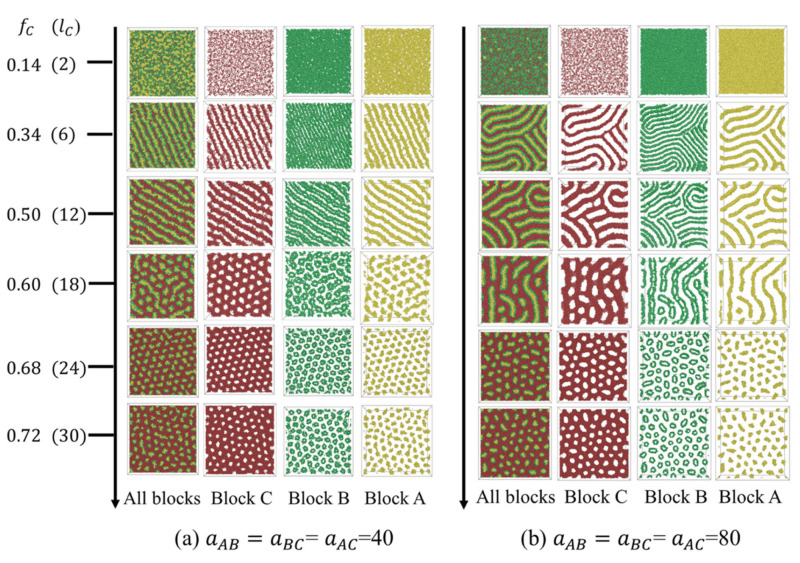
Phase transition sequence with $f_{C}$ along $f_{A}=f_{B}$ at different polymer–polymer interaction energy values. The morphology is shown in the xy plane.

**Figure 3 polymers-15-03982-f003:**
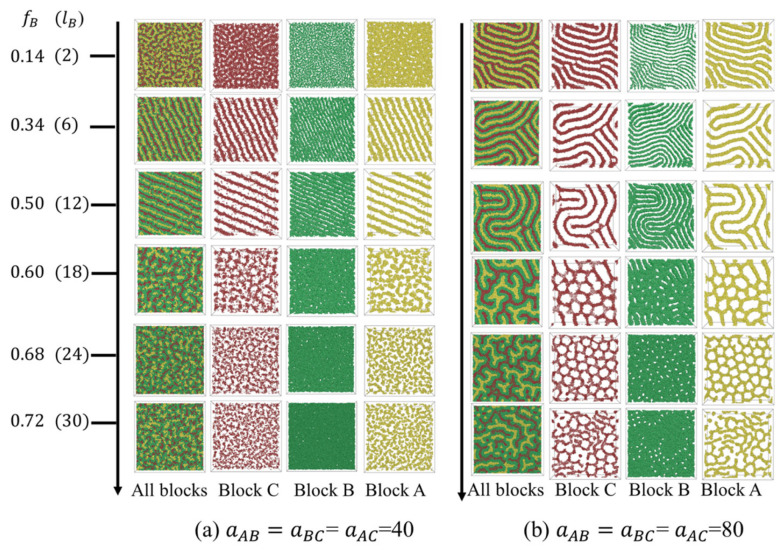
Phase transition sequence with $f_{B}$ along $f_{A}=f_{C}$ at different polymer–polymer interaction energy values. The morphology is shown in the xy plane.

**Figure 4 polymers-15-03982-f004:**
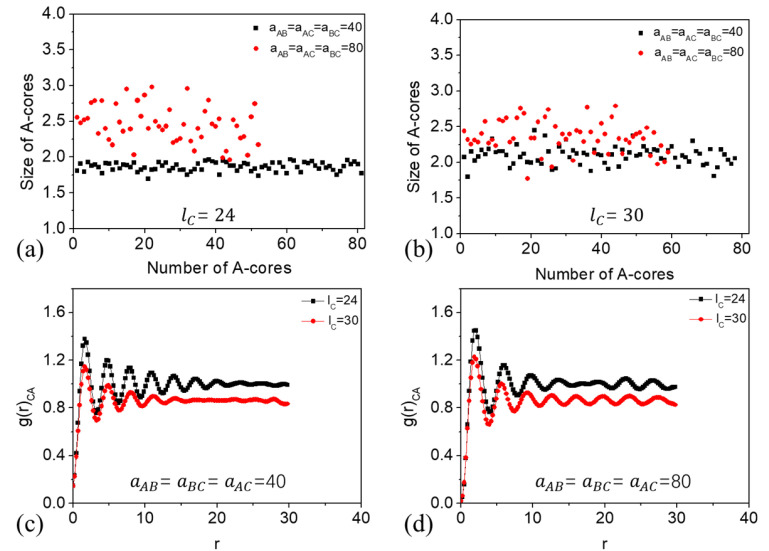
(**a**,**b**) Size of A-cores; (**b**–**d**) radial distribution function of components C and A under different C-block lengths and polymer–polymer interaction strengths.

**Figure 5 polymers-15-03982-f005:**
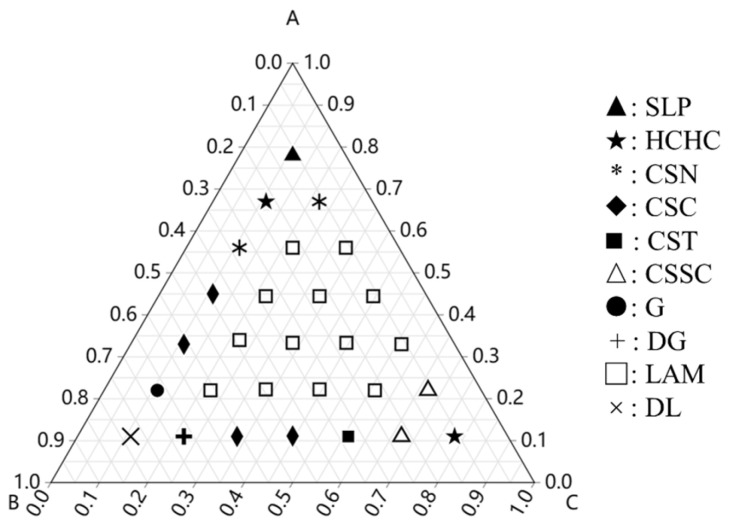
Phase triangle of CBABC linear pentablock terpolymers in terms of tree composition $f_{A}$, $f_{B}$, and $f_{C}$ at $a_{AB}=a_{BC}=a_{AC}=80$, $a_{PW}=120$.

**Figure 6 polymers-15-03982-f006:**
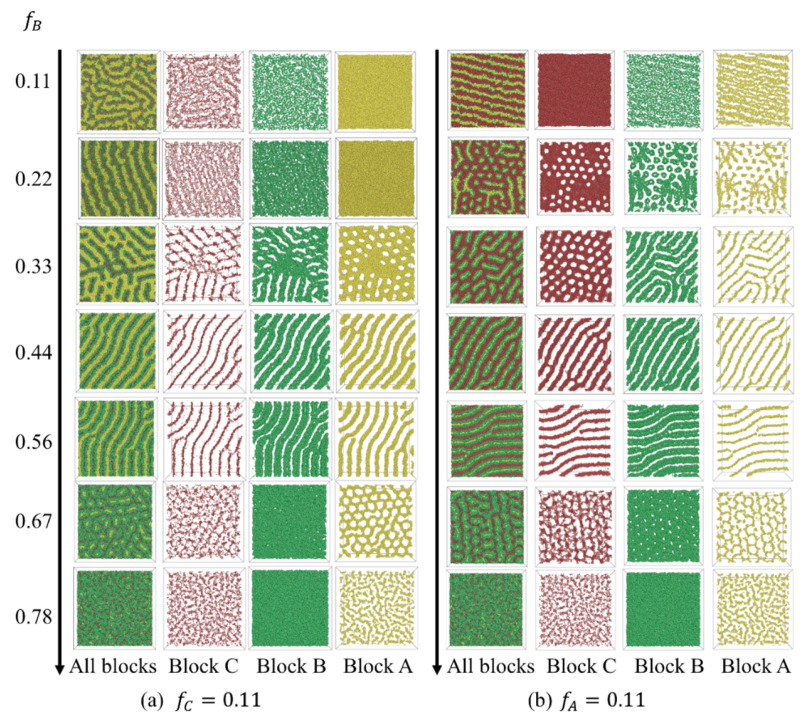
Phase transition sequence with $f_{B}$ fnd surface preference. at low $f_{C}$ and $f_{A}$. The morphology is shown in the xy plane.

**Figure 7 polymers-15-03982-f007:**
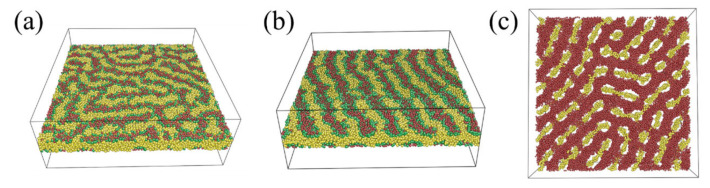
(**a**) Single layer with pattern (SLP); (**b**) half core–shell cylinder (HCSC); (**c**) core–shell thread structure (CST).

**Figure 8 polymers-15-03982-f008:**
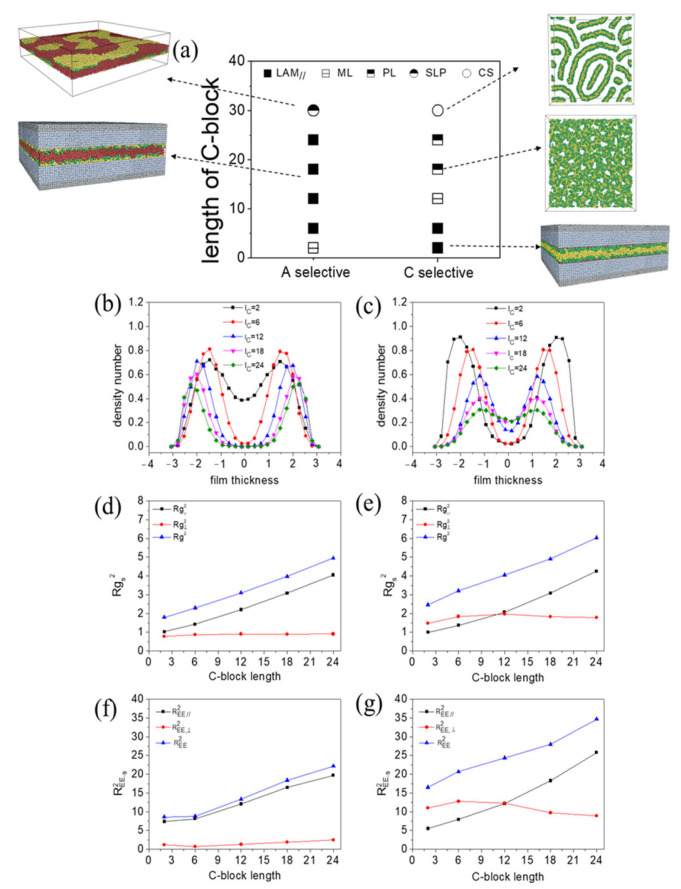
(**a**) Phase diagram of CBABC pentablock terpolymers showing the relationship between C-block length $l_{C}$ and surface preference. (**b**,**c**) Density number distribution profile for C-blocks across varying lengths $l_{C}$. (**d**,**e**) Mean square radius of gyration $R_{g}^{2}$ as a function of block length. $R_{g,∥}^{2}$ and $R_{g,⊥}^{2}$ represent the components of $R_{g}^{2}$ that are parallel and perpendicular to the wall, respectively. (**f**,**g**) Mean square end-to-end distance $R_{EE}^{2}$ as a function of block length. $R_{EE,∥}^{2}$ and $R_{EE,⊥}^{2}$ denote the components of $R_{EE}^{2}$ that are parallel and perpendicular to the wall, respectively.

**Figure 9 polymers-15-03982-f009:**
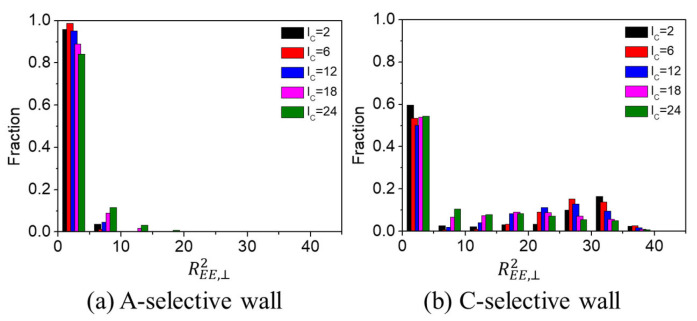
Histogram showing the perpendicular components $R_{EE,⊥}^{2}$ at various conditions.

**Figure 10 polymers-15-03982-f010:**
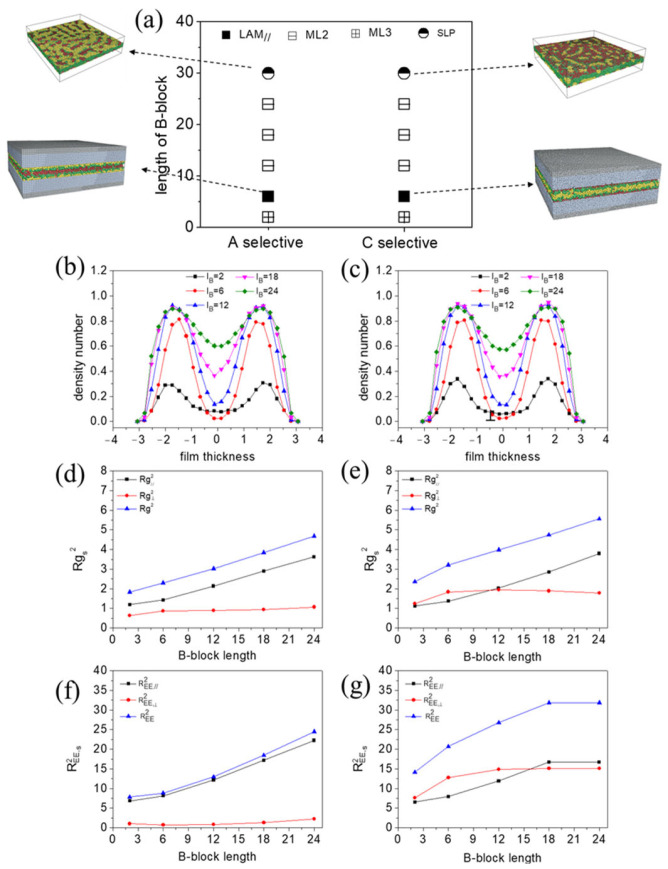
(**a**) Phase diagram of CBABC pentablock terpolymers showing the relationship between B-block length $l_{B}$ and surface preference. (**b**,**c**) Density number distribution profile for B-blocks across varying lengths $l_{B}$. (**d**,**e**) Mean square radius of gyration $R_{g}^{2}$ as a function of block length. $R_{g,∥}^{2}$ and $R_{g,⊥}^{2}$ represent the components of $R_{g}^{2}$ that are parallel and perpendicular to the wall, respectively. (**f**,**g**) Mean square end-to-end distance $R_{EE}^{2}$ as a function of block length. $R_{EE,∥}^{2}$ and $R_{EE,⊥}^{2}$ denote the components of $R_{EE}^{2}$ that are parallel and perpendicular to the wall, respectively.

**Table 1 polymers-15-03982-t001:** Conditions for the formation of typically observed self-assembled structures of CBABC pentablock terpolymers under ultra-confinement.

	Formation Condition		
Morphologies	Wall Properties	Block–Block Interaction Strength	Fraction of Blocks
Perpendicular lamellae	Nonselective wall	Not related	$f_{A}\approx f_{B}\approx f_{C}$
Parallel lamellae	A-selective wall	Weak	$f_{C}$ $=0.34\sim 0.68,\;f_{A}=f_{B}$
	C-selective wall	Weak	$f_{A}$ $=0.34\sim 0.68,\;f_{B}=f_{C}$
Perforated lamellae	Nonselective wall	Not related	$f_{A}$ = 0.6~0.72
			$f_{C}$ = 0.6~0.72
	A-selective wall	Weak	$f_{A}$ = 0.60~0.68
	C-selective wall	Weak	$f_{C}$ = 0.60~0.68
Cylinder	Nonselective wall	Weak	$f_{A}\approx 0.5,\;f_{B}=f_{C}$
		Weak	$f_{B}\approx 0.5,\;f_{A}=f_{C}$
		Weak	$f_{C}\approx 0.5,\;f_{A}=f_{B}$
Gyroid	Nonselective wall	Strong	$f_{B}$ $=0.6\sim 0.68,\;f_{A}=f_{C}$

## Data Availability

The data presented in this study are available on request from the corresponding author. The data are not publicly available due to privacy.

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
