# Peer review of "Dissipative Particle Dynamics Simulation for the Self-Assembly of Symmetric Pentablock Terpolymers Melts under 1D Confinements"

_polymers, 2023, doi:10.3390/polym15193982_

Round 1

Reviewer 1 Report

  1. Within the framework of the main question, the authors studied the phase behavior of CBABC pentabloc terpolymers embedded in thin films.
  2. In the current version of the article, everything is chaotic and it is difficult to judge originality. Many studies have been carried out in this direction (10.1016/j.eurpolymj.2017.10.008). Authors should better structure the article.
  3. The author focuses on issues related to the method of dissipative particle dynamics. Conducts analysis of structural and phase parameters of polymers.
  4. The authors must propose a hypothesis that will allow us to evaluate the influence of the structural properties of the polymer used on various macroproperties.
  5. The conclusions should be finalized, as they do not allow assessing the scientific novelty of the work in comparison with the works of other authors.
  6. Links are suitable, but they should be expanded - to reveal the essence of the research goal.
  7. It is necessary to add additional tables that will allow you to compare the results with the work of other researchers.

 Moderate editing of English language required

Author Response

Dear Reviewer,

The comments have been replied point-by-point. Please see the attachment.

Reviewer 2 Report

I cannot recommend publication for a manuscript which has just simulated results without confirmation by experimental parts in Polymers

Author Response

Dear Reviewer,

Reviewer 3 Report

In this work, the Authors studied the self-assembly behavior of pentablock terpolymers using Dissipative Particle Dynamic method. The Authors discussed the different morphologies influenced by the chain length, block composition and wall selectivity.

In general, the manuscript is well written and the obtained results are well presented. However, I think that the manuscript lacks an experimental part in which the described films are actually prepared and analyzed by imaging in order to confirm that the simulated structures are the ones observable in reality. This is important because in the self-assembly process, the interactions that are formed could be much more complex than the simulated ones.

Below, other minor observations are reported:

In the introduction section, I suggest to increase the number of references regarding the use of self-assembly structures used to tune the polymers properties. Among others, I recommend to the authors to see and cite these recent papers:

-       -   Polymers 2020, 12(10), 2183; DOI: 10.3390/polym12102183

-       -   ACS Appl. Polym. Mater. 2023, 5, 3, 1902–1914. DOI: 10.1021/acsapm.2c01972

In materials and methods section I suggest to describe better the chemical nature of the symmetric CBABC pentablock terpolymers used in this study. I also suggest to replace the schematic representation of the polymer in Figure 1 with the chemical structure of the A, B and C blocks used for the simulation analysis.

Author Response

(The authors gave the same response as above.)

Round 2

Reviewer 1 Report

 Accept in present form

Author Response

Thank you once again for your insightful and valuable feedback.

Reviewer 2 Report

Reject

Author Response

Thank you for your comments.  In future work, the author will conduct some experiments to validate the simulation results. 

Reviewer 3 Report

I understand that the work is purely simulative, however the experimental verification would have been interesting to add. I hope the authors follow the suggestion for a future article.
